# REVERSE DISTILLATION: CONSISTENTLY SCALING PROTEIN LANGUAGE MODEL REPRESENTATIONS

**Darius Catrina**[*]
Department of Computer Science
Duke University
Durham, NC, USA
`darius.catrina@duke.edu`

**Christian Bepler**[*]
Department of Computer Science
Duke University
Durham, NC, USA
`christian.bepler@duke.edu`

**Samuel Sledzieski**[†]
Center for Computational Biology
Flatiron Institute
New York, NY, USA
`ssledzieski@flatironinstitute.org`

**Rohit Singh**[†]
Depts of Biostats. and Bioinfo. & Cell Bio.
Duke University
Durham, NC, USA
`rohit.singh@duke.edu`

## ABSTRACT

Unlike the predictable scaling laws in natural language processing and computer vision, protein language models (PLMs) scale poorly: for many tasks, models within the same family plateau or even decrease in performance, with mid-sized models often outperforming the largest in the family. We introduce *Reverse Distillation*, a principled framework that decomposes large PLM representations into orthogonal subspaces guided by smaller models of the same family. The resulting embeddings have a nested, Matryoshka-style structure: the first $k$ dimensions of a larger model's embedding are exactly the representation from the smaller model. This ensures that larger reverse-distilled models consistently outperform smaller ones. A motivating intuition is that smaller models, constrained by capacity, preferentially encode broadly-shared protein features. Reverse distillation isolates these shared features and orthogonally extracts additional contributions from larger models, preventing interference between the two. On ProteinGym benchmarks, reverse-distilled ESM-2 variants outperform their respective baselines at the same embedding dimensionality, with the reverse-distilled 15 billion parameter model achieving the strongest performance. Our framework is generalizable to any model family where scaling challenges persist. Code and trained models are available at
`https://github.com/rohitsinghlab/plm_reverse_distillation`.

## 1 INTRODUCTION

Protein language models (PLMs) have emerged as powerful representation learners, capturing evolutionary patterns from millions of sequences and enabling unprecedented capabilities in structure prediction (Lin et al., 2023), function annotation (Yu et al., 2023), and protein design (Ferruz et al., 2022; Devkota et al., 2024). These models learn rich protein representations through self-supervised training on vast sequence databases. However, unlike the predictable scaling laws observed in natural language processing (Kaplan et al., 2020; Hoffmann et al., 2022), PLMs—and more broadly, biological foundation models—exhibit counterintuitive scaling behavior: smaller models often outperform larger ones on functional prediction tasks (Li et al., 2024). For example, on deep mutational scanning (DMS) benchmarks from ProteinGym (Notin et al., 2023), the ESM-2 family peaks at 650M-3B parameters, with the 15B model showing degraded performance.

This unexpected scaling behavior creates fundamental challenges. Given models $M_1, M_2$ with $|M_2| > |M_1|$ parameters, we observe non-monotonic performance: we often find that smaller models outperform larger models on downstream tasks. Moreover, we cannot reliably predict which

---

[*]Equal contribution.
[†]Co-corresponding authors.

biological tasks will exhibit poor scaling behavior, leading to difficulties in model selection for any specific task. A related limitation of PLMs is that embeddings across model scales are disconnected. In contrast, Matryoshka-style embeddings (Kusupati et al., 2022) in natural language processing are structured such that their prefixes are also directly usable. With these embeddings, prefixes of an overall embedding are themselves functional, albeit with some performance degradation. This enhances computational and storage efficiency, enabling a paradigm of "embed once, reuse prefixes as needed.". However, current PLM representations do not offer this advantage: representations of dimension $k$ cannot be truncated to dimension $k' < k$ while maintaining smooth performance degradation.

A motivating intuition for our approach comes from the bias-variance tradeoff. Small PLMs, constrained by capacity, are biased toward encoding frequent, broadly-shared biological regularities—secondary structure propensities, hydrophobicity patterns, or conserved structural motifs. Larger models have the capacity to additionally represent rarer, higher-order phenomena: family-specific patterns, epistatic interactions, allosteric signals. But this additional capacity also introduces variance: when higher-order features are entangled with simpler ones in a single representational space, downstream linear predictors struggle to isolate task-relevant signal. The result is that task-irrelevant features effectively add noise to the fundamental patterns that drive most benchmark performance.

We introduce **Reverse Distillation**[*], a principled framework that decomposes large PLM representations into orthogonal subspaces anchored by smaller models. Unlike traditional knowledge distillation, which compresses large models into small ones, our method identifies the unique information contributed by each model scale. Our key insight is structural: by treating smaller model representations as a basis and extracting orthogonal residuals from the larger model, we prevent destructive interference between features at different scales. In doing so, reverse distillation introduces a Matryoshka-style relationship between embeddings of different model sizes, enabling predictable scaling behavior as model size increases.

Formally, given models $M_r$ and $M_p$ where $|M_r| < |M_p|$, with embedding dimensionality $k_r < k_p$, we decompose representations as $H_p \approx [H_r, H_{res}]$ where $H_r \in \mathbb{R}^{n \times k_r}$ captures more fundamental patterns learned by the smaller model and $H_{res} \in \mathbb{R}^{n \times (k_p - k_r)}$ captures unique information from the larger model, orthogonal to $H_r$. We prove that this decomposition is MSE-optimal among all $k_p$-dim representations that fully encompass $M_r$ (i.e., the cylinder set of $M_r$ in $\mathbb{R}^{k_p}$). Our contributions are:

- **Hierarchical Decomposition**: We show how to transform a family of PLM models such that they follow a hierarchical structure where each higher scale adds orthogonal information. Our decomposition is also guaranteed to approximate the original representation space well.

- **Matryoshka-style Embeddings and Monotonic Improvement**: Reverse-distilled embeddings are constructed such that nested embeddings of dimensionality $d$ contain prefixes of sizes $d_1 < d_2 < d_3 < d$ such that each prefix is the reverse-distilled embedding of the corresponding dimensionality. Our decomposition thus provides controlled performance degradation as a function of embedding size.

- **Scaling Consistency**: Reverse distillation scales nearly always, i.e., larger reverse-distilled models consistently perform better than smaller ones.

- **Improvement over Baseline**: For the ESM-2 family, reverse-distilled models of the same embedding size (e.g., 1280 for ESM-2 650M) generally outperform their corresponding baselines.

## 2 METHOD

**Motivation and Intuition** The ESM-2 family spans embedding dimensions from 320 (8M parameters) to 5120 (15B parameters), providing a systematic testbed for analyzing PLM scaling behavior. Each model learns residue co-evolution patterns through self-attention mechanisms, but capacity constraints induce different feature distributions across model scales.

---

[*]The term "reverse distillation" has appeared in certain teacher-student architectures in ML literature. Our usage—decomposing large models using smaller ones as a basis—is distinct and we believe intuitive from context.

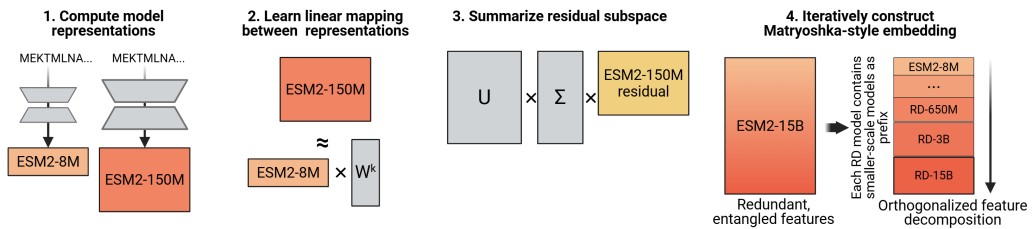

Figure 1: **Overview of Reverse Distillation** Large protein language models (e.g., ESM-2$_{3B}$) entangle representations of diverse features in a single representational space, hindering the performance of downstream linear probes. Reverse distillation constructs a product space by preserving the smaller model's representation (capturing more conserved features) and extracting orthogonal residuals via SVD (capturing features unique to the larger model). Iterating this process across a model family yields Matryoshka-style embeddings where each prefix corresponds to a valid reverse-distilled representation at that scale. Figure created using biorender.io

Small models operate under severe capacity constraints. To minimize loss at training time, these models must prioritize the most frequently occurring co-evolutionary patterns—those that maximize compression across the protein sequence distribution. These patterns likely correspond to general protein properties: secondary structure propensities, hydrophobicity patterns, and conserved structural motifs. The smallest ESM-2 model (8M parameters) lacks sufficient capacity to represent rarer patterns; its limited parameters are allocated to features with the highest marginal utility across the training distribution.

Large models possess capacity to represent more diverse functional features. They can encode enzyme-specific catalytic motifs, protein family-specific allosteric couplings, and higher-order interaction patterns. However, as Li et al. (2024) demonstrated, downstream performance often relies on early, low-level features rather than additional capacity; consequently, linear probes on larger, mixed representations frequently fail to isolate task-relevant signal from task-irrelevant variance.

Our key intuition is that smaller models provide a natural basis for decomposition (Fig. 1). A smaller model from the same family—trained on identical data with the same architecture—produces representations biased toward broadly-shared features due to capacity constraints. By computing the orthogonal complement of the smaller model's subspace within the larger model's representation, we achieve partial separation of features without requiring explicit feature identification.

We employ linear decomposition throughout our framework to maintain interpretability. Linear methods reveal directly accessible information in representations without confounding from nonlinear prediction heads, and enable precise attribution of information to specific model scales. While nonlinear methods might achieve higher downstream performance, linear decomposition provides the analytical tractability necessary to characterize how biological information distributes across the model hierarchy.

## 2.1 PROBLEM FORMULATION

Consider a hierarchy of protein language models $\mathcal{M} = \{M_1, M_2, \ldots, M_m\}$ ordered by parameter count. Each model $M_i$ maps sequences to embeddings:

$$M_i : \mathcal{P}^* \rightarrow \mathbb{R}^{n \times k_i}$$

where $\mathcal{P}$ is the amino acid alphabet, $n$ varies per sequence, and $k_1 < k_2 < \ldots < k_m$ are embedding dimensions. For the ESM-2 family, this hierarchy exists naturally, with model scales at 8M ($k_1 = 320$), 35M ($k_2 = 640$), 150M ($k_3 = 480$), 650M ($k_4 = 1280$), 3B ($k_5 = 2560$), and 15B ($k_6 = 5120$) parameters.

While our framework does not require monotonically increasing dimensions—appropriate dimensionality reduction via PCA or variational autoencoders could enable reverse distillation between arbitrary model pairs—the ESM-2 family's architecture provides this structure directly, simplifying our implementation.

For a protein sequence dataset $\mathcal{D} = \{s_i\}_{i=1}^N$ with sequence lengths $\{n_i\}$, the total amino acid positions $L = \sum_{i=1}^N n_i$ represents our effective sample size for learning linear relationships. This formulation (treating all positions as samples) enables data-efficient subspace learning. For training, we used $N = 1,000$ sequences sampled randomly from UniRef50; all sequences had 30% or lower

sequence identity to datasets used in Section 3. Due to the simple linear transforms involved in this work, we found this $N$ to be sufficient.

**Reverse Distillation Decomposition**

**Definition 1** (Reverse Distillation Decomposition). *Given models $M_r$ and $M_p$ where $r < p$, we decompose the representation space $\mathbb{R}^{n \times k_p}$ into orthogonal subspaces:*

$$\mathcal{S}_p = \mathcal{S}_r \oplus \mathcal{S}_{res}$$

*where $\mathcal{S}_r \cong \mathbb{R}^{n \times k_r}$ preserves $M_r$'s representations and $\mathcal{S}_{res} \cong \mathbb{R}^{n \times (k_p - k_r)}$ captures orthogonal residual information.*

We express any representation $H_p \in \mathbb{R}^{n \times k_p}$ as:

$$H_p \approx [H_r, H_{res}]$$

where $H_r \in \mathbb{R}^{n \times k_r}$ comes directly from the smaller model $M_r$, and $H_{res} \in \mathbb{R}^{n \times (k_p - k_r)}$ represents the unique contribution of the larger model.

We preserve entire smaller models rather than selecting subsets of their dimensions. This choice, enabled by the natural progression of embedding sizes ($320 \to 640 \to 1280 \to 2560 \to 5120$ in ESM-2), maintains interpretability—we know $H_r$ represents the complete feature space learned by the smaller model, making the residual space $\mathcal{S}_{res}$ directly interpretable as features unique to the larger model.

## 2.2 ALGORITHMS

Algorithm 1 presents our training procedure. Phase 1 comprises forward passes of the $M_r$ and $M_p$, computing representations of the same sequences at different scales.

---

**Algorithm 1** Reverse Distillation Algorithm (Pre-training)

---

**Require:** Dataset $\mathcal{D} = \{s_i\}_{i=1}^N$ where $|s_i| = n_i$, models $M_r$, $M_p$ with $r < p$
**Ensure:** Subspace decomposition matrices $\mathbf{W}^*$, $\mathbf{V}_{res}$
 1: **Phase 1: Compute Representations**
 2: **for** $i = 1$ to $N$ **do**
 3:    $H_r^{(i)} = M_r(s_i) \in \mathbb{R}^{n_i \times k_r}$ {Variable length $n_i$}
 4:    $H_p^{(i)} = M_p(s_i) \in \mathbb{R}^{n_i \times k_p}$
 5: **end for**
 6: **Phase 2: Learn Linear Mappings**
 7: Define total length: $L = \sum_{i=1}^N n_i$
 8: Stack representations: $\tilde{H}_r = \text{vstack}(H_r^{(1)}, \ldots, H_r^{(N)}) \in \mathbb{R}^{L \times k_r}$
 9: Stack representations: $\tilde{H}_p = \text{vstack}(H_p^{(1)}, \ldots, H_p^{(N)}) \in \mathbb{R}^{L \times k_p}$
10: Solve: $\mathbf{W}^* = \arg\min_{\mathbf{W}} \|\tilde{H}_p - \tilde{H}_r \mathbf{W}\|_F^2$    ▷ Via prin. comp. reg. (PCR); noise PCs of $\tilde{H}_r$ removed
11: Compute residuals: $\mathbf{R} = \tilde{H}_p - \tilde{H}_r \mathbf{W}^* \in \mathbb{R}^{L \times k_p}$
12: **Phase 3: Subspace Identification**
13: Apply SVD: $\mathbf{R} = \mathbf{U}\boldsymbol{\Sigma}\mathbf{V}^T$
14: Select top $(k_p - k_r)$ components: $\mathbf{V}_{res} = \mathbf{V}[:, 1 : (k_p - k_r)]$
15: **return** $\mathbf{W}^*$, $\mathbf{V}_{res}$

---

**Refinements to the linear mapping.** In Phase 2 of Algorithm 1, we use principal component regression (PCR) rather than ordinary least squares. Since some dimensions of $\tilde{H}_r$ are likely noise, we project onto its principal components and apply the Johnstone threshold (Johnstone, 2001) from random matrix theory to discard likely-noise components before regression (Appendix A.1). We also verified that reverse distillation performs comparably to a stronger but non-Matryoshka baseline: concatenating all model scales and reducing via PCA (Appendix A.2).

Algorithm 2 shows inference. The decomposed representation $H_{rd} = [H_r, H_{res}]$ is Matryoshka by construction—prefixes correspond to valid smaller model outputs, enabling adaptive compute at deployment.

Algorithm 3 extends reverse distillation from a pair of models to a whole family of models. Chaining reveals hierarchical structure where each scale contributes orthogonal information that cannot be linearly predicted from smaller models.

**Theoretical Analysis** Let $\mathcal{M}_r \subset \mathbb{R}^{k_r}$ be the manifold spanned by embeddings of $M_r$. To enable scalability and flexibility in our representation space, it is desirable to enforce the Matryoshka property on our embeddings. Thus, we consider the set of all $k_p$-dimensional representations that preserve $M_r$'s embeddings in their first $k_r$ coordinates:

$$\mathcal{C}_r = \{[H_r, X] : H_r \in \mathcal{M}_r, X \in \mathbb{R}^{L \times (k_p - k_r)}\}$$

Our decomposition $H_{rd} = [H_r, H_{res}]$ minimizes reconstruction error within this constrained space:

**Theorem 1** (Optimal Constrained Approximation). *Let $\tilde{H}_p \in \mathbb{R}^{L \times k_p}$ and $\tilde{H}_r \in \mathbb{R}^{L \times k_r}$ be stacked representations from models $M_p$ and $M_r$ respectively, where $r < p$. Among all representations of the form $[\tilde{H}_r, X]$ where $X \in \mathbb{R}^{L \times (k_p - k_r)}$, the representation $H_{rd} = [\tilde{H}_r, H_{res}]$ with $H_{res}$ derived from the top $(k_p - k_r)$ singular vectors of the residual $\mathbf{R} = \tilde{H}_p - \tilde{H}_r \mathbf{W}^*$ minimizes:*

$$\min_{A \in \mathbb{R}^{k_p \times k_p}} \|\tilde{H}_p - [\tilde{H}_r, X]A\|_F^2$$

*Proof.* The proof directly follows from the Eckart-Young theorem. The optimal linear predictor is $\mathbf{W}^* = (\tilde{H}_r^T \tilde{H}_r)^{-1} \tilde{H}_r^T \tilde{H}_p$, minimizing the reconstruction error over all $\mathbf{W} \in \mathbb{R}^{k_r \times k_p}$. The residual $\mathbf{R} = \tilde{H}_p - \tilde{H}_r \mathbf{W}^*$ contains information orthogonal to $\tilde{H}_r$. For $\mathbf{R} = \mathbf{U}\mathbf{\Sigma}\mathbf{V}^T$, the optimal rank-$(k_p - k_r)$ approximation uses the top $(k_p - k_r)$ singular vectors. $\square$

## 3 EXPERIMENTS

### 3.1 INITIAL EXPLORATION OF MODEL CHAIN CONFIGURATION

We began by investigating the optimal chaining of small models into larger models. For three DMS datasets from ProteinGym (Notin et al., 2023), we evaluated a range of chain configurations. Let $K = \{k_0, k_1, \ldots, k_n\}$ denote the $n + 1$ model sizes from a model family (for ESM-2: $K = \{8M, 35M, 150M, 650M, 3B, 15B\}$). For a target embedding $k_t$ with $t \in [0, n]$, the chain configuration was defined as follows:

1. for each $k_i \in [0, n]$ with $i < t$, a direct chain $k_i \rightarrow k_t$
2. longest chain: $k_0 \rightarrow k_1 \rightarrow \cdots \rightarrow k_t$

As shown in Table 1, a consistent trend emerged in which longer incremental chains yielded improved performance. Consequently, we concentrated our comprehensive experiments on the results obtained from reverse distillation of the two largest models.

In the rest of the paper, we denote the chain $k_{8M} \rightarrow \cdots \rightarrow k_{650M}$ as **rd.650**, the chain $k_{8M} \rightarrow \cdots \rightarrow k_{3B}$ as **rd.3B**, and the chain $k_{8M} \rightarrow \cdots \rightarrow k_{15B}$ as **rd.15B**.

### 3.2 PROTEINGYM DMS ANALYSIS

For a comprehensive analysis, we obtained datasets from ProteinGym with at least one double- or multi-mutation variant. To ensure that our evaluation estimates were reliable, we excluded datasets with fewer than 100 single-mutation variants. Given an embedding scheme, for each protein in the dataset, we loaded the embedding of the wild-type sequence and the embeddings of the mutated sequence. For each mutation, we computed the difference vector between the embeddings of the mutated sequence and the corresponding wild-type sequence at the mutated position, feeding it into a ridge regression classifier. For variants with multiple mutations, we first average the differences

---

**Algorithm 2** Reverse Distillation Inference

**Require:** New sequence $s$ with $|s| = n$, learned matrices $\mathbf{W}^*$, $\mathbf{V}_{res}$, models $M_r$, $M_p$
**Ensure:** Decomposed representation $H_{rd} \in \mathbb{R}^{n \times k_p}$
1: $H_r = M_r(s) \in \mathbb{R}^{n \times k_r}$ {Smaller model embedding}
2: $H_p = M_p(s) \in \mathbb{R}^{n \times k_p}$ {Larger model embedding}
3: $H_{pred} = H_r \mathbf{W}^* \in \mathbb{R}^{n \times k_p}$ {Predicted large model embedding}
4: $R = H_p - H_{pred} \in \mathbb{R}^{n \times k_p}$ {Unexplained residuals}
5: $H_{res} = R\mathbf{V}_{res} \in \mathbb{R}^{n \times (k_p - k_r)}$ {Projected residuals}
6: $H_{rd} = [H_r, H_{res}] \in \mathbb{R}^{n \times k_p}$ {Concatenate reference + residual}
7: **return** $H_{rd}$

---

---

**Algorithm 3** Chained Reverse Distillation

---

**Require:** Dataset $\mathcal{D}$, model hierarchy $\{M_1, \ldots, M_m\}$
**Ensure:** Decomposition components $\{\mathbf{W}_i, \mathbf{V}_i\}_{i=2}^m$

1: Initialize: $H_{acc}^{(1)} = M_1(\mathcal{D})$, $k_{acc}^{(1)} = k_1$
2: **for** $i = 2$ to $m$ **do**
3:     $H_i = M_i(\mathcal{D})$
4:     Learn predictor: $\mathbf{W}_i = \arg\min_{\mathbf{W}} \|H_i - H_{acc}^{(i-1)}\mathbf{W}\|_F^2$
5:     Compute residuals: $R_i = H_i - H_{acc}^{(i-1)}\mathbf{W}_i$
6:     Apply SVD: $R_i = \mathbf{U}_i\mathbf{\Sigma}_i\mathbf{V}_i^T$
7:     Select components: $\mathbf{V}_i = \mathbf{V}_i[:, 1 : (k_i - k_{acc}^{(i-1)})]$
8:     Update: $H_{acc}^{(i)} = [H_{acc}^{(i-1)}, R_i\mathbf{V}_i]$
9:     Update: $k_{acc}^{(i)} = k_i$
10: **end for**
11: **return** $\{\mathbf{W}_i, \mathbf{V}_i\}_{i=2}^m$

---

Table 1: **Progressive chain vs. direct chain.** Our approach supports reverse distillation of any smaller model into any larger model. However, we find empirically that the best performance comes from progressively distilling up a "chain" of models. In the remainder of this paper, we refer to these progressive chains as rd.650M, rd.3B, and rd.15B.

| ESM Models | ARGR_ECOLI
Tsuboyama_2023_1AOY | DN7A_SACS2
Tsuboyama_2023_1JIC | ILF3_HUMAN
Tsuboyama_2023_2L33 |
|---|---|---|---|
| 8M | 0.771 | 0.746 | 0.670 |
| 35M | 0.767 | 0.806 | 0.692 |
| rd: 8M→35M | 0.776 | 0.793 | 0.701 |
| 150M | 0.799 | 0.786 | 0.760 |
| rd: 35M→150M | 0.811 | 0.791 | 0.772 |
| rd: 8M→150M | 0.820 | 0.792 | 0.779 |
| 650M | 0.834 | 0.868 | 0.712 |
| rd: 8M→650M | 0.849 | 0.878 | 0.765 |
| rd: 35M→650M | 0.835 | **0.881** | 0.759 |
| rd: 150M→650M | 0.845 | 0.866 | 0.751 |
| rd: 8M→35→150→650M (rd.650M) | **0.858** | 0.867 | **0.786** |
| 3B | 0.845 | 0.880 | 0.749 |
| rd: 8M→3B | 0.852 | **0.898** | 0.780 |
| rd: 35M→3B | 0.844 | 0.894 | 0.777 |
| rd: 150M→3B | 0.853 | 0.886 | 0.775 |
| rd: 650M→3B | 0.859 | 0.880 | 0.751 |
| rd: 8→35→150→650→3B (rd.3B) | **0.873** | 0.890 | **0.801** |

across all mutated positions. We fit the ridge regression on 80% of the single-mutational variants using leave-one-out cross-validation. The trained model was used to predict for all multiple-mutants and the remaining single-mutant cases. Note that since rd.650M (rd.3B) is a prefix of rd.3B (rd.15B) by construction (the first 1280 (2560) dimensions are identical), any cases where rd.650M (rd.3B) outperforms rd.3B (rd.15B) likely reflect ridge regression artifacts rather than representational limitations. For each DMS dataset, we computed the Spearman correlation between our predicted scores and the ground truth; we followed the original ProteinGym manuscript in our choice of the Spearman metric. We report an estimated per-dataset measure of improvement, asking "in how many datasets does model $M_1$ outperfom $M_2$?" (Table 2) along with the mean and standard deviations of these correlations for both the ESM-2 family of models and their reverse-distilled version (Table 3).

## 3.3 ADDITIONAL PROTEIN PROPERTY PREDICTION

We evaluated our reverse-distilled models on several downstream protein property prediction tasks where the prediction task directly corresponded to protein structural, functional, and dynamic features. We utilized the 3-class secondary structure prediction (SSP Q3), 8-class secondary structure prediction (SSP Q8), metal ion binding (MIB) and localization prediction (LOC) benchmarks from Chen et al. (2024), as well as the R2/R1 prediction task from Wayment-Steele et al. (2025). Training

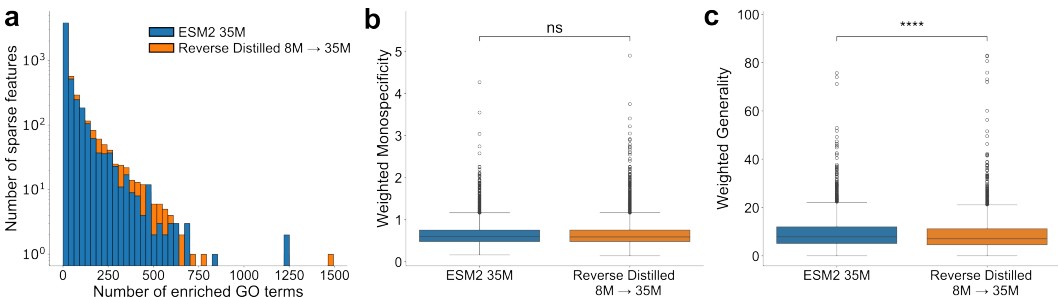

Figure 2: **Reverse Distillation embeddings capture more GO terms.** **(a)** SAE features from the rd.35M model are enriched for more GO terms than those from the base model, indicating that they contain more functionally relevant information. **(b)** The sets of GO terms for each model are equally compact, measured by pairwise shortest path on the GO tree. **(c)** The sets of GO terms for rd.35M SAE features are significantly less general, measured by the depth of the pairwise least common ancestor on the GO tree. While features from the base model capture high-level GO terms, reverse distillation enables model features to represent distinct functions.

was performed analogously to the previous setting. The reverse-distilled models frequently outperformed the base models (Table 4) and demonstrated consistent scaling, with rd.15B nearly always outperforming rd.3B and the baseline ESM models.

## 3.4 PROBING EMBEDDINGS WITH SPARSE AUTOENCODERS

Finally, we sought to explore whether our reverse-distilled embeddings could disentangling representations of biological features by training sparse autoencoders (SAE). We followed the analysis of Gujral et al. (2025), training an SAE on embeddings from both the base ESM-2 35M and our rd.35M model. Using a set of 18,142 proteins from the UniProt database, we identified the proteins significantly associated with each sparse feature, then performed a GO enrichment analysis using annotations for the proteins (Ashburner et al., 2000) to identify the set of GO terms associated with each sparse feature. We found that the SAE trained on rd.35M embeddings contained more enriched GO terms (40 GO terms for the average rd.35M SAE feature, vs. 32 for the average ESM-2 35M SAE feature) indicating that these embeddings capture more functional features than those from the base model (Figure 2a).

In addition, the DAG relationship between GO terms allows us to probe the relationships within each functional set. Following Gujral et al. (2025) we compute "monospecificity," the inverse of average shortest pairwise distance between GO terms in the same set (Figure 2b), and "generality," the depth of the least common ancestor (LCA) of each pair of terms in the set (Figure 2c). A high monospecificity means that SAE features capture a more consistent set of functional features, while a high generality means that SAE features capture less specific biological function. While rd.35M and ESM-2 35M embeddings have similar levels of monospecificity, rd.35M embeddings are significantly less general, lending support to our hypothesis that reverse distillation helps to disentangle representations of biological features within the model.

Table 2: **Reverse Distillation restores scaling on ProteinGym benchmarks.** We show that not only do rd.650M, rd.3B, and rd.15B consistently outperform their baseline counterparts, but that reverse distillation preserves expected scaling, i.e. larger models more frequently outperform smaller models.

| | # ProteinGym DMS Datasets | % ProteinGym DMS Datasets where one model outperforms another | | | | | | |
|---|---|---|---|---|---|---|---|---|
| | | rd 650M > 650M | rd 3B > 3B | rd 15B > 15B | 3B > 650M | 15B > 3B | rd 3B > rd 650M | rd 15B > rd 3B |
| **ProteinGym DMS Datasets with 1 and 2 mutations** | | | | | | | | |
| 1 mut | 28 | 71.43% | 85.71% | 67.86% | 53.57% | 92.86% | 92.86% | 85.71% |
| 2 mut | 28 | 53.57% | 53.57% | 57.14% | 46.43% | 85.71% | 67.86% | 75.00% |
| **ProteinGym DMS Datasets with >2 mutations** | | | | | | | | |
| 1 mut | 6 | 83.33% | 16.67% | 66.67% | 100.00% | 33.33% | 100.00% | 66.67% |
| 2 mut | 6 | 33.33% | 66.66% | 83.33% | 33.33% | 83.33% | 100.00% | 100.00% |
| 3 mut | 6 | 66.66% | 66.66% | 66.66% | 66.67% | 83.33% | 100.00% | 100.00% |
| 4 mut | 6 | 50.00% | 50.00% | 50.00% | 50.00% | 66.67% | 100.00% | 100.00% |

Table 3: **Spearman correlation of predicted mutational effect on ProteinGym benchmarks.** We report the performance of a classifier trained on embeddings from various models to predict variant effect on deep mutational scanning (DMS) data from ProteinGym. rd.15B achieves the strongest performance out of any model tested, and the reverse distilled models generally outperform their baseline counterparts.

| | # ProteinGym DMS Datasets | Test Spearman correlation | | | | | |
| --- | --- | --- | --- | --- | --- | --- | --- |
| | | (mean ± std) | | | | | |
| | | 650M | rd 650M | 3B | rd 3B | 15B | rd 15B |
| **ProteinGym DMS Datasets with 1 and 2 mutations** | | | | | | | |
| 1 mut | 28 | 0.879 (± 0.040) | **0.885 (± 0.038)** | 0.881 (± 0.043) | **0.893 (± 0.039)** | 0.899 (± 0.036) | **0.904 (± 0.037)** |
| 2 mut | 28 | 0.672 (± 0.140) | **0.688 (± 0.133)** | 0.672 (± 0.128) | **0.697 (± 0.115)** | 0.714 (± 0.115) | **0.720 (± 0.120)** |
| **ProteinGym DMS Datasets with >2 mutations** | | | | | | | |
| 1 mut | 6 | 0.534 (± 0.192) | **0.550 (± 0.187)** | **0.591 (± 0.189)** | 0.574 (± 0.165) | 0.579 (± 0.180) | **0.580 (± 0.189)** |
| 2 mut | 6 | **0.608 (± 0.131)** | 0.600 (± 0.137) | 0.621 (± 0.146) | **0.643 (± 0.132)** | 0.653 (± 0.156) | **0.677 (± 0.134)** |
| 3 mut | 6 | **0.578 (± 0.056)** | 0.577 (± 0.068) | 0.583 (± 0.103) | **0.617 (± 0.082)** | 0.607 (± 0.183) | **0.652 (± 0.084)** |
| 4 mut | 6 | **0.555 (± 0.067)** | 0.539 (± 0.089) | 0.552 (± 0.145) | **0.576 (± 0.090)** | 0.566 (± 0.171) | **0.615 (± 0.086)** |

## 3.5 Inference time

On an Nvidia A6000 GPU, embedding a protein sequence (mean length = 536) took 0.09s and 0.249s for the ESM-2 650M and 3B models respectively. Even though rd.650M involves four ESM model-invocations (8M, 35M, 150M and 650M) it only took 1.69x(=0.152s) the time as the smaller models have faster inference. Similarly, rd.3B makes five model-invocations but took only 1.53x (=0.380s) the time compared to baseline ESM-2 3B, and the six model-invocations of rd.15B take only 1.70x as long as the base model. Thus, reverse distillation does not have a prohibitive inference overhead. We also note that the prefix structure of the embeddings enhances reusability.

## 4 Related Work

**The PLM scaling problem is well-documented.** The ProteinGym benchmark shows that performance gains plateau around 1–4B parameters, with hybrids leveraging MSAs or structure often winning on zero-shot fitness (Notin, 2025). Multiple lines of evidence reinforce this ceiling: improvements with scale are largely task-dependent, with linear probes on large representations struggling to isolate task-relevant signal (Li et al., 2024); the ESM-2 3B model delivers contact-recovery signals comparable to the 15B variant (Zhang et al., 2024); medium-sized models perform competitively in realistic transfer settings (Vieira et al., 2025); and variant-effect accuracy peaks at intermediate model perplexity, with both under- and over-confident models degrading discrimination (Hou et al., 2025). Complementary work on representational redundancy shows that PLM embeddings can be substantially compressed along both sequence and feature dimensions (Lu et al., 2025; Devkota et al., 2024), suggesting that large models devote significant capacity to redundant or entangled features. Collectively, these findings characterize the scaling problem; reverse distillation offers a solution by systematically decomposing representations across scales.

**Relationship to distillation, model combination, and continual learning.** Traditional knowledge distillation (Hinton et al., 2015) compresses a large teacher into a smaller student; model

Table 4: **Reverse distillation of ESM-2 improves performance on downstream protein property prediction tasks.** We evaluate ESM-2 650M, 3B and 15B and their corresponding reverse-distilled versions on four data sets from Chen et al. (2024): 3-class secondary structure prediction (SSP Q3), 8-class secondary structure prediction (SSP Q8), metal ion binding (MIB), and localization prediction (LOC)) We also report on R2/R1 prediction from Wayment-Steele et al. (2025). Across nearly all data sets, we find that rd.15B achieves the strongest performance.

| Dataset | 650M | rd 650M | 3B | rd 3B | 15B | rd 15B |
| --- | --- | --- | --- | --- | --- | --- |
| SSP Q3 (*aupr* ↑) | 0.831 | 0.833 | 0.791 | 0.816 | 0.845 | **0.861** |
| SSP Q8 (*aupr* ↑) | 0.365 | 0.369 | 0.379 | 0.395 | 0.418 | **0.431** |
| MIB (*aupr* ↑) | 0.881 | 0.855 | 0.893 | 0.891 | 0.900 | **0.901** |
| LOC (*aupr* ↑) | 0.709 | 0.669 | 0.705 | 0.708 | **0.745** | 0.743 |
| R2/R1 (*aupr* ↑) | 0.343 | 0.405 | 0.369 | 0.425 | 0.368 | **0.468** |

soups (Wortsman et al., 2022) average weights across fine-tuned variants. Both aim to consolidate information into a single model. Reverse distillation instead decomposes representations across model scales, identifying each scale's unique contributions. Our use of orthogonal subspaces is related to continual learning methods like o-LoRA (Wang et al., 2023) and Adaptive SVD (Nayak et al., 2025), which maintain orthogonal weight subspaces when adapting to new tasks. However, these methods are task-specific; reverse distillation is task-agnostic, extracting representations from a multi-scale model family by maximizing residual information gain.

**Relationship to interpretability methods.** Attention-head analyses ("BERTology") probe what individual heads encode (Rogers et al., 2020; Clark et al., 2019; Vig et al., 2021), while sparse autoencoders (SAEs) enumerate latent features explicitly (Gujral et al., 2025; Adams et al., 2025). Both inspect representations but neither decomposes them across model scales. Moreover, mapping SAE latents to biologically relevant features demands extensive manual annotation. Reverse distillation sidesteps this by operating implicitly—separating what different model scales contribute without pre-defining or cataloging individual features.

## 5 Conclusion

The success of reverse distillation suggests that scaling challenges in PLMs stem from inefficient use of representational capacity rather than fundamental limits in model expressiveness. A purely linear decomposition—requiring no model retraining overhead—restores monotonic scaling and improves over baselines at the same embedding dimensionality. This indicates that the information needed for consistent scaling is already present in large models; the challenge is extracting it. By reframing the question from "when do large models help?" to "how can we systematically combine contributions across scales?", reverse distillation opens new avenues for representation analysis and more effective scaling strategies.

**Limitations and Future Work.** Our current framework relies on linear decomposition, which provides theoretical guarantees and interpretability but may leave nonlinear relationships between model scales unexploited. Initial explorations of nonlinear mappings show improved reconstruction ($8M \rightarrow 35M$ $R^2 = 0.528$ vs. $0.422$; $650M \rightarrow 3B$ $R^2 = 0.400$ vs. $0.261$), suggesting room to further extract unique features from larger models. Nonlinear dimensionality reduction (e.g., UMAP (McInnes et al., 2018)) could also more effectively separate residual features, and would be particularly valuable for model families where different-sized models share the same embedding dimension.

A complementary direction is to use parameter-efficient fine-tuning (e.g., LoRA) to produce reverse-distilled embeddings directly at the last layer of a large model. This would enable generative use-cases and likelihood-based probing while requiring only a single forward pass, eliminating the current multi-model inference overhead.

Finally, we plan to test reverse distillation on other biological foundation models—including autoregressive PLMs like ProGen, as well as models for genomics and drug discovery—and on foundation models outside biology. We anticipate that the core principle of leveraging smaller models to effectively decompose larger ones may apply broadly wherever scaling challenges persist.

### Acknowledgments

R.S. acknowledges the support of the Whitehead Scholarship at the Duke University School of Medicine. D.C. acknowledges the support of the Karsh International Scholars Program from Duke University. S.S. acknowledges support from the Simons Foundation.

## 6 Reproducibility Statement

To facilitate easy verification and replication of our results, we have modularized our source code, added README documentation and provided information on how to train the reverse distillation models. Code is available on GitHub at `https://github.com/rohitsinghlab/plm_reverse_distillation`, and pre-trained models are available to download via the package.

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

## A  APPENDIX

### A.1  ABLATION ON LINEAR MAPPING DURING REVERSE DISTILLATION (PHASE 2 OF ALGORITHM 1)

To map the smaller model embeddings $H_r$ to the larger model embeddings $H_p$, we explored two approaches: a) an ordinary least-squares regression (OLS), and b) principle-component regression (PCR). Since there may be "noise" features in $H_r$, we hypothesized that using PCR would let us utilize only the informative features of $H_r$ for predicting $H_p$. Towards that, we first mapped $H_r$ to its principal components, and then applied principles from random matrix theory (specifically, the Johnstone threshold (Johnstone, 2001)) to select the top $r_j$ likely-non-noise PCs, and used only these to predict $H_p$. We compare the performance of PCR and OLS in Figure 3, showing that on PCR consistently yields stronger downstream performance across multiple mutation data sets.

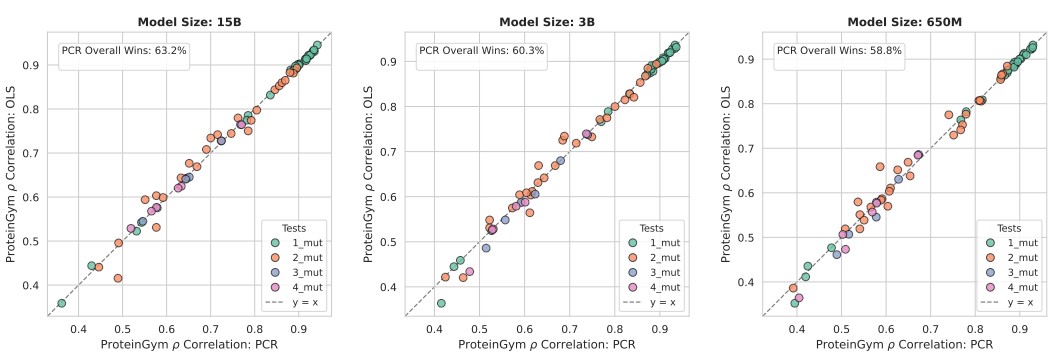

Figure 3: **Ablation on Linear Mapping** The plots show the performance of the rd.650M, rd.3B, and rd.15B models using both PCR and OLS and evaluated on their Spearman Correlation ($\rho$) across 37 datasets from the Protein Gym benchmark. They show that PCR consistently outperforms OLS, with overall win rates of 58.8%, 60.3%, and 63.2% respectively. This suggests that isolating signal from noise in the lower-scale representations is critical for effective cross-scale distillation

### A.2  ABLATION ON SCALE-PRESERVING FEATURE CONCATENATION

An alternative to our Matryoshka approach—while still preserving a linear subspace-decomposition approach—is to simply concatenate embeddings of models and then perform dimensionality reduction via principal component anlaysis (PCA). Thus, to distill embeddings $H_r$ into $H_p$, we would concatenate the two, perform PCA, and take the top $p$ components. In a sense, this represents the upper bound of representational orthogonality achievable by any linearly-constructed $p$-dimensional subspace that invokes all model scales up to $H_p$. At the same time, these representations are *not* Matryoshka-style: when graduating to a higher model scale, the resulting embeddings (since they will be produced by a fresh PCA) may be entirely different. As we show in Figure A.2, our reverse distillation representations performs similarly to this representation while preserving the Matryoshka property.

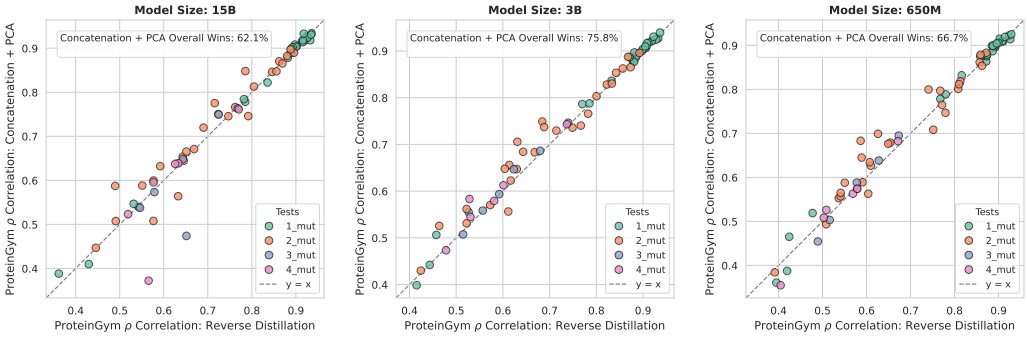

Figure 4: **Ablation on Matryoshka property**: **(Top)** Comparison of Spearman correlation $\rho$ on ProteinGym datasets between our Reverse Distillation approach and a PCA + concatenation baseline. While the baseline provides a theoretical upper bound for linear subspace decomposition, **(Bottom)** highlights its critical limitation: unlike our method, it relies on fresh projections at each scale and therefore cannot preserve structural consistency or scalability across the 650M, 3B, and 15B models.

| | **Naive PCA + concatenation scalability on ProteinGym DMS Datasets** | | | |
|---|---|---|---|---|
| | **naive 3B > naive 650M** | **naive 15B > naive 3B** | **rd 3B > rd 650M** | **rd 15B > rd 3B** |
| **ProteinGym DMS Datasets with 1 and 2 mutations** | | | | |
| 1 mut | 85.71% | 71.43% | 92.86% | 85.71% |
| 2 mut | 76.19% | 57.14% | 67.86% | 75.00% |
| **ProteinGym DMS Datasets with >2 mutations** | | | | |
| 1 mut | 85.71% | 14.29% | 100.00% | 100.00% |
| 2 mut | 85.71% | 71.43% | 100.00% | 66.67% |
| 3 mut | 85.71% | 71.43% | 100.00% | 100.00% |
| 4 mut | 85.71% | 71.43% | 100.00% | 100.00% |

