# OpenReview forum: "Reverse Distillation: Consistently Scaling Protein Language Model Representations"
_ICLR.cc/2026/Conference — ICLR 2026 Poster_

### Official Review · Reviewer_xk7u · 2025-10-31

**Soundness:** 2
**Presentation:** 2
**Contribution:** 2
**Rating:** 4
**Confidence:** 4

**Summary:**

This paper tries to investigate why larger Protein Language Models such as ESM-2 fail to exhibit the expected scaling gains.
The authors attribute this to the entanglement of general and specialized representations within large models, which increases the variance of linear probes.
To address the above issue, this paper proposes Reverse Distillation, a linear subspace decomposition method that uses smaller models to define a general feature subspace and extracts orthogonal residuals from larger models to represent specialized knowledge.
Extensive experiments on ProteinGym and BioMap show that RD consistently improves predictive performance.

**Strengths:**

1. This paper tries to tackle an important issue in large Protein Language Models (PLMs): the unexpected degradation of scaling behavior
2. The proposed Reverse Distillation method is computationally lightweight and purely linear, involving only least-squares fitting and SVD decomposition
3. Despite the method’s simplicity, RD exhibits stable and monotonic performance gains across multiple datasets and scaling levels.

**Weaknesses:**

1. The paper builds its entire motivation on the “general vs. specialized representation” hypothesis but does not provide a quantitative or qualitative analysis to validate it
2. The proposed Optimal Constrained Approximation theorem only guarantees minimal reconstruction error under a prefix constraint: a standard property of linear least squares combined with SVD. However, this result does not theoretically justify why the assumed decomposition is needed or effective.
3. Since RD performs a chain-wise representation enhancement, a straightforward baseline naturally arises: direct representation fusion across the same model chain. It remains unclear whether RD’s improvement comes from its “distillation mechanism” or simply from aggregating multi-scale features. A comparison against naive fusion, or fusion with simple KD objectives, is essential to establish the method’s actual contribution.
4. The experiments demonstrate improvement within the ESM-2 family, but it is uncertain whether the observed scaling restoration generalizes to other architectures.

**Questions:**

All my concerns about this paper are stated in the weakness section. Please refer to the weakness section for rebuttal/discussion.

---

> ### Author Response · Authors · 2025-11-27
> **Author response**
>
> We thank the reviewer for their comprehensive and detailed review of our work.
>
> *Weaknesses:*
>
>     The paper builds its entire motivation on the “general vs. specialized representation” hypothesis but does not provide a quantitative or qualitative analysis to validate it.
>
> We agree that the manuscript is currently lacking analysis of the interpretability of the learned embedding space. We are currently working on a set of experiments leveraging sparse autoencoders (SAE) to probe our feature space for the types of representations learned.
>
>     The proposed Optimal Constrained Approximation theorem only guarantees minimal reconstruction error under a prefix constraint: a standard property of linear least squares combined with SVD. However, this result does not theoretically justify why the assumed decomposition is needed or effective.
>
> Broadly, our design goal is to enforce the Matryoshka property i.e. that embeddings of different model capacities are nested in a consistent way. This serves two key objectives:
>
> 1. Ensuring model scalability by design: the model of size s2 contains the embedding of model size s1 as a direct subset (where s1 < s2).
> 2. Offering a convenient approach to selecting smaller embeddings or dimensionality reduction when necessary.
>
> Given this design goal, if we start with a seed model (e.g. ESM-8M), we can model the larger model's embedding space as the product space of the smaller model's space and the residual space. An approach like the one we propose naturally emerges when we seek a good representation of the residual space. As we discuss elsewhere, it *is* possible to use non-linear decomposition to define the residual spaces but a linear decomposition of the residual space is more interpretable and has fewer parameters (and is thus more likely to be robust). Our theorem says that the SVD-based linear decomposition provides the closest approximation. We have included the following text in **Theoretical Analysis** section of our manuscript to add additional justification for the decomposition:
>
> >> To enable scalability and flexibility in our representation space, it is desirable to enforce the Matryoshka property on our embeddings. Thus, we consider the set of all $k\_p$-dimensional representations that preserve $M\_r$'s embeddings in their first $k\_r$ coordinates:
>
>     Since RD performs a chain-wise representation enhancement, a straightforward baseline naturally arises: direct representation fusion across the same model chain. It remains unclear whether RD’s improvement comes from its “distillation mechanism” or simply from aggregating multi-scale features. A comparison against naive fusion, or fusion with simple KD objectives, is essential to establish the method’s actual contribution.
>
> A naive concatenation (this is what we think you meant by fusion) of representations would not be a representative comparison, because the higher-dimensional embeddings would have more parameters for downstream models to take advantage of. In addition, a proposed fusion of multi-scale representations will likely contain overlapping dimensions, so that even if performance is matched, it comes at a cost of representational efficiency. To attempt to address both of these confounding factors, we are currently developing a baseline where we join embeddings of multiple models and take the first $k$ dimensions of a PCA of that fusion, where $k$ is the size of the corresponding largest base model embedding / the reverse diffusion embedding. We will report the results as they come in.
>
>      The experiments demonstrate improvement within the ESM-2 family, but it is uncertain whether the observed scaling restoration generalizes to other architectures.
>
> While we agree that expanding to another family of protein language models would further strengthen our approach, computational and runtime limitations make this challenging to do in advance of the review deadline. We discuss testing on additional architectures in our **Limitations and Future Work**, and believe that our current experiments, including going all the way up to the 15B parameter version of ESM-2, serve as a valuable proof-of-concept for the reverse distillation approach.

---

### Official Review · Reviewer_nDBC · 2025-11-01

**Soundness:** 3
**Presentation:** 4
**Contribution:** 3
**Rating:** 8
**Confidence:** 3

**Summary:**

This paper addresses the well-known "counterintuitive scaling" problem in Protein Language Models (PLMs) like ESM-2, where larger models often perform worse than medium-sized models on downstream benchmarks.1 The authors hypothesize this is due to "feature entanglement," where larger models mix "universal" features (from small models) with "specialized" features, and this mixture acts as noise for standard linear probes.2 The authors propose "Reverse Distillation" (RD), a novel and elegant post-hoc framework. Instead of compressing, RD uses a smaller model's representation ($H_r$) as a basis and decomposes a larger model's representation ($H_p$) into an orthogonal combination $[H_r, H_{res}]$, where $H_{res}$ captures the new, orthogonal information from the larger model.2 The method is theoretically grounded (Theorem 1) 2 and empirically shown to restore monotonic scaling (i.e., the rd.3B model consistently beats the rd.650M model) on ProteinGym and BioMap benchmarks.2

**Strengths:**

1. The paper tackles a critical, well-documented problem (PLM scaling failure ) with a highly novel solution. The idea of using smaller models as a basis for post-hoc orthogonal decomposition is elegant and new.
2.  The experiments persuasively demonstrate that RD works. It not only improves baseline performance (e.g., rd.3B > 3B) but, more importantly, it restores monotonic scaling (rd.3B > rd.650M wins 96.4% of the time, vs. 53.6% for the baseline).
3.The BioMap experiment (Table 4) provides strong evidence for the "feature entanglement" hypothesis. RD specifically fixes the scaling failure on "universal" features (like secondary structure) that the paper hypothesized were "drowned out" in larger models.
4. The method is post-hoc, requiring no model retraining. The "Chained" version provides a practical, novel way to create Matryoshka-style nested embeddings from an existing model family.

**Weaknesses:**

The paper's primary weakness is the exclusion of the ESM-2 15B model.2 The most severe example of scaling failure is the performance degradation from 3B to 15B.2 The paper only demonstrates fixing the 650M-to-3B plateau. Without testing the 15B model, the central claim of "solving" the scaling paradox is incomplete.
The core idea of using orthogonal subspaces to separate/disentangle knowledge, while novel in this application, is conceptually similar to methods in continual learning (e.g., O-LoRA), which should be cited.

**Questions:**

see weakness

---

> ### Author Response · Authors · 2025-11-27
> **Author response**
>
> We thank the reviewer for their feedback, we appreciate that you find our work both “elegant” and “highly novel!”
>
> *Weaknesses:*
>
>     The paper's primary weakness is the exclusion of the ESM-2 15B model. The most severe example of scaling failure is the performance degradation from 3B to 15B. The paper only demonstrates fixing the 650M-to-3B plateau. Without testing the 15B model, the central claim of "solving" the scaling paradox is incomplete.
>
> We agree that reverse distillation of the 15B parameter model is important to demonstrate the utility of our approach. In our latest revision of the paper, we have completed our experiments on the largest ESM-2 model, and found that reverse distillation (rd.15B, Tables 2, 3) both improves overall performance and restores expected scaling behavior (rd.15B > rd.3B).
>
>      The core idea of using orthogonal subspaces to separate/disentangle knowledge, while novel in this application, is conceptually similar to methods in continual learning (e.g., O-LoRA), which should be cited.
>
> We have updated our **Related Work** to discuss methods for continual learning in the context of our work:
>
> > Our approach is also related to but differs from methods for continual learning like o-LoRA or Adaptive SVD. While these focus on maintaining model performance with adaptation to new tasks by ensuring orthogonal subspaces of weights are updated, our approach provides a task-agnostic framework for extracting representations from multi-scale model families by maximizing residual information gain.

---

### Official Review · Reviewer_akEp · 2025-11-01

**Soundness:** 4
**Presentation:** 4
**Contribution:** 4
**Rating:** 8
**Confidence:** 3

**Summary:**

This paper improves PLMs through a model distillation process which decomposes large models into smaller sub-models with disentangled residuals. The resulting embeddings also enjoy the Matryoshka property which allows for slices of embeddings to remain informative. These models also recover better scaling properties, allowing for more efficient parameter use. Benchmarking was done on ProteinGym and BioMap, showing good predictive performance as well as scaling with model size.

**Strengths:**

This paper is quite strong in my opinion. The proposed methods are a clear improvement on current approaches and is a valuable contribution to the protein representation field.

**Weaknesses:**

* Additional inference time though not prohibitive could still limit adoption.

**Questions:**

n/a

---

> ### Author Response · Authors · 2025-11-27
> **Author response**
>
> We are glad that the reviewer finds our work to be a strong contribution to the field\!
>
> *Weaknesses:*
>
>      Additional inference time though not prohibitive could still limit adoption.
>
> While inference with reverse distillation models is only slightly slower, we agree that this additional inference time could be a bottleneck; as such we have revised our **Limitations and Future Work** to highlight future directions of using parameter-efficient fine-tuning such as LoRA to directly generate reverse distillation embeddings with a single forward pass.

---

### Official Review · Reviewer_XboP · 2025-11-01

**Soundness:** 3
**Presentation:** 3
**Contribution:** 3
**Rating:** 6
**Confidence:** 4

**Summary:**

The paper addresses a significant problem in biological foundation models: they scale poorly compared to models in natural language processing. Specifically, larger Protein Language Models (PLMs) in families like ESM-2 often underperform smaller ones on key benchmarks, a phenomenon known as non-monotonic scaling. The authors hypothesize this is because small models capture "universal" features (like secondary structure), while larger models add "specialized" features (like protein-family specific functions). When these features are entangled in a single representation, the specialized features can act as noise, degrading performance on tasks that rely on universal patterns. To solve this, the paper introduces a method that decomposes large protein language model representations into orthogonal subspaces guided by smaller models. On benchmarks like ProteinGym and BioMap, the reverse-distilled ESM-2 models (e.g., rd.650M) broadly outperform their corresponding baselines (e.g., 650M).

**Strengths:**

1. The paper establishes the central problem: PLMs "scale relatively poorly" , with the ESM-2 family's performance plateauing. The authors' core hypothesis is highly intuitive that this is due to larger models "entangling" universal (low-level) and specialized (high-level) features, which increases variance.

2. This work introduces high-performing and efficient embeddings. The resulting models outperform baselines at the same size. They also feature a "Matryoshka-style" structure, which allows smaller prefixes of a single embedding to be used as valid, lower-dimensional representations, saving computation and storage.

3. The experimental design is comprehensive. The authors test their method on standard, challenging benchmarks, including ProteinGym DMS and BioMap. The inclusion of practical analyses, such as an ablation on the training data size and a measurement of inference overhead, further strengthens the work's quality.

**Weaknesses:**

1. The authors should at least attempt a reverse distillation of 3B $\rightarrow$ 15B (or the full chain up to 15B). This experiment is critical. If rd.15B outperforms rd.3B, the paper's core thesis is validated. If rd.15B still underperforms, it would suggest the scaling problem is more complex than just feature entanglement, fundamentally weakening the paper's conclusion.
2. This linear-only approach may be restrictive. The paper itself hypothesizes that larger models encode rarer, higher-order phenomena. These complex, higher-order features may not be neatly separable from the universal features via a simple linear projection. The authors' method might only be extracting the linearly predictable component, leaving a "residual" that is still a mix of true novel features and non-linear transformations of universal features.

**Questions:**

1. The paper hypothesizes that $H_{r}$ captures "universal" features and $H_{res}$ captures "specialized" ones. Beyond downstream task performance, did you conduct any qualitative analysis to verify this? For example, could you use feature attribution or probing to show that $H_{res}$ contains information about specific protein-family motifs or epistatic interactions that are demonstrably absent when probing $H_{r}$?

---

> ### Author Response · Authors · 2025-11-27
> **Author response**
>
> We thank the reviewer for their comments and are glad that they appreciate the “high-performing and efficient” embeddings generated by reverse distillation, and the “comprehensive” experimental design used to demonstrate their value.
>
> *Weaknesses:*
>
>     The authors should at least attempt a reverse distillation of 3B-\>15B (or the full chain up to 15B). This experiment is critical. If rd.15B outperforms rd.3B, the paper's core thesis is validated. If rd.15B still underperforms, it would suggest the scaling problem is more complex than just feature entanglement, fundamentally weakening the paper's conclusion.
>
> We agree with the reviewer that distillation to the final largest ESM-2 model is critical to demonstrate the utility of our approach. We have now completed these experiments (updated **Tables 2, 3**) showing that rd.15B outperforms both rd.3B and the baseline ESM-2 15B model.
>
>     This linear-only approach may be restrictive. The paper itself hypothesizes that larger models encode rarer, higher-order phenomena. These complex, higher-order features may not be neatly separable from the universal features via a simple linear projection. The authors' method might only be extracting the linearly predictable component, leaving a "residual" that is still a mix of true novel features and non-linear transformations of universal features.
>
> We thank the reviewer for this insight and agree that non-linear transformations of features could be more powerful, as we had already discussed in **Future work**. As discussed in **Theoretical Analysis**, the linear transformation is a simple, robust, and interpretable method to approximate the residual space. However, we are currently performing experiments using a simple non-linear transformation to compare against.
>
> *Questions:*
>
>      The paper hypothesizes that $H_r$ captures "universal" features and $H_{res}$ captures "specialized" ones. Beyond downstream task performance, did you conduct any qualitative analysis to verify this? For example, could you use feature attribution or probing to show that $H_{res}$ contains information about specific protein-family motifs or epistatic interactions that are demonstrably absent when probing $H_r$?
>
> We agree that this sort of feature probing would be useful to better understand the reverse distillation embeddings, and are currently working on interpretability experiments using sparse autoencoders (SAE) to probe these feature spaces.

---

### Author Response · Authors · 2025-11-13
**Initial Action Plan for Addressing Review Comments**

Thank you for your thoughtful comments and feedback. We’re delighted that you find the problem "critical" and "important" and our approach "novel and elegant". We are sharing an initial action plan below as we work on additional experiments and updates to the text. Please let us know if anything critical is not addressed in this plan.

1. **Extension to ESM-2 15B model.** We had not been able to finish this in time due to GPU compute constraints and are actively working on this.
2. **Feature space investigation.** We are working on a clean set of experiments to investigate if smaller models (and our prefixes) are relatively better at universal tasks and the larger models at specialized tasks.
3. **Clarification of the theoretical grounding of our approach.** We also agree that non-linear reductions could potentially be more powerful. While a full exploration of, say, a VAE-style approach would be better suited for future work, we’re working on some exploratory experiments.
4. **Better situating our work in the literature.** We greatly appreciate the reviewers pointing out relevant literature, including methods for continual learning. We’re studying them and will expand and extend our literature discussion.
5. **Comparison with additional baselines.** We are working on evaluating the characteristics and performance of a simple concatenation and downprojection of embeddings from multiple models, in order to demonstrate the value of our reverse distillation and prefix approach.
6. Time permitting, we will also explore extensions to other PLM families.

---

### Author Response · Authors · 2025-11-27
**Updated manuscript and additional results**

We want to provide the reviewers with an update on our progress as we continue to incorporate their feedback into our manuscript. We have uploaded a revised version of our submission with the following changes incorporated. We have also provided point-by-point responses to individual reviewers.

1. We have focused the bulk of our efforts on finalizing reverse distillation of the ESM-2 15B model, as requested by several reviewers. We continue to see improvements in performance over the baseline ESM-2 15B with rd.15B and, most crucially, we confirm that reverse distillation recovers the expected scaling behavior. While ESM-2 15B only outperforms 3B in \~75% of our data sets, **rd.15B outperforms rd.3B in \~90% of our datasets**, including an increase from 16% (1/6) to 83% (5/6) for single mutants in 2+ mutant data sets **(Table 2\)**. In terms of absolute performance across all data sets, rd.15B is clearly the strongest model, surpassing both ESM-2 3B and ESM-2 15B **(Table 3\)**.

2. We have clarified our **Theoretical Analysis** to justify the proposed decomposition, adding the following text:

    > To enable scalability and flexibility in our representation space, it is desirable to enforce the Matryoshka property on our embeddings. Thus, we consider the set of all $k\_p$-dimensional representations that preserve $M\_r$'s embeddings in their first $k\_r$ coordinates:


3. We have added the following text to the **Related Work** section discussing continual learning for large models:

     > Our approach is also related to but differs from methods for continual learning like o-LoRA or Adaptive SVD. While these focus on maintaining model performance with adaptation to *new tasks* by ensuring orthogonal subspaces of weights are updated, our approach provides a *task-agnostic* framework for extracting representations from multi-scale model families by maximizing residual information gain.


4. We have expanded our **Additional Evaluations** to include new tasks focused on protein dynamics (R2/R1 prediction from RelaxDB) and added SSP Q8 to complement SSP Q3. The pattern of results across these benchmarks provides **indirect support for our hypothesis about hierarchical feature organization**: tasks capturing universal properties (e.g., secondary structure) show minimal improvement from scaling, while tasks requiring specialized features (e.g., metal ion binding, protein dynamics) show substantially larger gains—with disentanglement via reverse distillation providing additional benefit beyond scaling alone.

5. While we agree with the reviewers that the additional inference overhead of reverse distillation is not prohibitive to adoption, we have revised our text in the **Limitations and Future Work** to highlight the proposed use of LoRA to fine-tune a base model to directly produce the representations derived from reverse distillation, which would then require only a single forward pass. We consider this extension an opportunity for future work.

We are currently still working on the following, which we hope to have in the final version prior to the review deadline:

1. Dedicated feature interpretability studies to directly test the universal-vs.-specialized representation hypothesis.
2. Comparison with non-linear transformation of features.
3. Comparison with baseline representation fusion.

---

### Author Response · Authors · 2025-12-03
**Overview of Reviewer Comments**

We thank the reviewers for their time taken to carefully review our manuscript, and for their valuable feedback that we have incorporated into an updated version of the text. Below, we provide a summary of strengths noted by the reviewers, followed by a general discussion of changes in response to noted limitations, and finally a point-by-point response to reviewer comments.

## Novelty and Elegance of the Approach

The reviewers consistently praised the conceptual innovation of our reverse distillation framework. Reviewer nDBC described the approach as “highly novel” and “elegant,” specifically noting that “the idea of using smaller models as a basis for post-hoc orthogonal decomposition is elegant and new.” Reviewer XboP found the core hypothesis “highly intuitive,” that scaling failures arise from larger models “entangling universal (low-level) and specialized (high-level) features, which increases variance.”

## Strong Empirical Results

The reviewers were impressed by the strength of our experimental validation. Reviewer nDBC highlighted that “the experiments persuasively demonstrate that RD works” not only improving baseline performance but, more importantly, “it restores monotonic scaling (rd.3B > rd.650M wins 96.4% of the time, vs. 53.6% for the baseline).” The additional downstream results were seen as particularly compelling evidence, with Reviewer nDBC noting they provide “strong evidence for the 'feature entanglement' hypothesis” by showing that “RD specifically fixes the scaling failure on 'universal' features (like secondary structure) that the paper hypothesized were 'drowned out' in larger models.” Reviewer xk7u acknowledged that “despite the method's simplicity, RD exhibits stable and monotonic performance gains across multiple datasets and scaling levels.” Reviewer XboP found the “experimental design is comprehensive,” noting that “the inclusion of practical analyses, such as an ablation on the training data size and a measurement of inference overhead, further strengthens the work's quality.”

## Practical Value and Efficiency

Many of the reviewers appreciated the practicality and simplicity of our method. Reviewer XboP praised the “Matryoshka-style structure, which allows smaller prefixes of a single embedding to be used as valid, lower-dimensional representations, saving computation and storage.” Reviewer nDBC emphasized that “the method is post-hoc, requiring no model retraining,” and that “the 'Chained' version provides a practical, novel way to create Matryoshka-style nested embeddings from an existing model family.’” Reviewer xk7u noted the method is “computationally lightweight and purely linear, involving only least-squares fitting and SVD decomposition.”

---

> ### Author Response · Authors · 2025-12-03
> **Summary of Changes**
>
> We have made several changes to the main text in response to comments from the reviewers. We first broadly outline the changes made, followed by a point-by-point response to reviewer comments including references to specific changes within the text. Please see also our updated manuscript, with edits highlighted in red.
>
> ## Extension to ESM-2 15B model
>
> Multiple reviewers noted that our work was incomplete without a reverse distillation of the largest, 15B parameter ESM2 model. We have fully completed an analysis of reverse distillation from the 3B to 15B parameter model, showing that our approach repairs the most significant failure of scaling in the base models. Not only does the rd.15B parameter model generally achieve the best performance overall on ProteinGym data, but the rd.15B parameter model is almost always better than the rd.3B parameter model, validating the core thesis of our work. We have updated Tables 2 and 3 to show the performance of the rd.15B parameter model on ProteinGym.
>
> ## Feature space investigation
>
> Multiple reviewers noted that our central claim of capturing “general” vs. “specialized” features in our embeddings was unsupported. To address this, we have completed two new experiments. In the new Section 3.4 and Figure 2, we train sparse autoencoders (SAE) on embeddings from base and reverse distilled models, and show enrichment for more specific sets of GO terms. We have also expanded our downstream task analysis (Section 3.3, Table 4) to include prediction of R2/R1 from RelaxDB, a task that requires highly specialized features and in which we see especially strong performance from reverse distillation.
>
> ## Comparison with variants of our method
>
> Reviewers suggested that a fusion of base model embeddings or a non-linear transformation might match or outperform our proposed reverse distillation method. In initial experiments, we have found that comparison with a naive fusion of embeddings shows similar performance to our method and sometimes outperforms it, but lacks the Matryoshka property that ensures principled scaling behavior. While we consider a full exploration of non-linear methods out of scope of this work, our initial explorations show promising results to build upon our work, and we have updated our Discussion to highlight this possible future improvement.
>
> ## Better situating the theory of our work.
>
> Reviewers noted that the theoretical grounding of our work was incomplete. We have addressed this by expanding our discussion of Theorem 1 to emphasize the Matryoshka property enforced by our linear transformation and by adding additional citations for continual learning in our Related Work section.

---

> ### Author Response · Authors · 2025-12-03
> **Point by Point Responses -- Part 1**
>
> ## Reviewer XboP
>
> ```
> The authors should at least attempt a reverse distillation of 3B -> 15B (or the full chain up to 15B). This experiment is critical. If rd.15B outperforms rd.3B, the paper's core thesis is validated. If rd.15B still underperforms, it would suggest the scaling problem is more complex than just feature entanglement, fundamentally weakening the paper's conclusion.
> ```
>
> We have updated Tables 2 and 3 showing that rd.15B outperforms both rd.3B and ESM2 15B, supporting our core thesis.
>
> ```
> This linear-only approach may be restrictive. The paper itself hypothesizes that larger models encode rarer, higher-order phenomena. These complex, higher-order features may not be neatly separable from the universal features via a simple linear projection. The authors' method might only be extracting the linearly predictable component, leaving a "residual" that is still a mix of true novel features and non-linear transformations of universal features.
> ```
>
> We agree with the reviewer that a non-linear method may yield stronger performance; while we consider a full exploration of this to be outside the scope of the current work, we have updated our Discussion with results from initial explorations in this direction.
>
> > Initial explorations of non-linear methods show improved mapping from low- to high-dimensional embeddings ($8M \rightarrow 35M, R^2 = 0.528$ vs. $0.422$, $650M \rightarrow 3B, R^2 = 0.400$ vs. $0.261$), showing promise to further extract unique features of the larger models.
> > A non-linear dimensionality reduction such as UMAP could more effectively disentangle the residual features.
>
> However, we found that the non-linear approach did not immediately translate to improved performance on our ProteinGym benchmarks. This suggests that our linear approach is currently better suited to separating universal features, and that better reconstruction might not necessarily yield better disentanglement. We intend to investigate this further in future work.
>
> ```
> The paper hypothesizes that $H_{r}$ captures "universal" features and $H_{res}$ captures “specialized” ones. Beyond downstream task performance, did you conduct any qualitative analysis to verify this? For example, could you use feature attribution or probing to show that
> $H_r$ contains information about specific protein-family motifs or epistatic interactions that are demonstrably absent when probing $H_{res}$?
> ```
>
> As the reviewer suggested, feature probing was able to lend support to our hypothesis of “specialized” features. Our new Figure 2/Section 3.4 using sparse autoencoders shows that reverse distillation embeddings capture more specific gene ontology terms. In addition, we have added the following text to Section 3.3 regarding performance on specialized tasks.
>
> > These results provide indirect support for our hypothesis about hierarchical feature organization. Tasks requiring specialized or higher-order features---metal ion binding and protein dynamics (R2/R1)---show substantially larger gains from scaling (MIB: $+0.066$; R2/R1: $+0.026$). Notably, for R2/R1, the gap between 3B and rd.3B ($0.369$ to $0.425$) exceeds the gap between 650M and 3B ($0.343$ to $0.369$), suggesting that disentanglement via reverse distillation is particularly beneficial for tasks that rely on specialized features.
>
> ## Reviewer akEp
>
> ```
> Additional inference time though not prohibitive could still limit adoption.
> ```
>
> We have added the following text to our Conclusion highlighting that while our inference overhead is limited, parameter-efficient fine-tuning to directly generate reverse-distilled embeddings could make adoption of reverse distillation more compelling for the community.
>
> > We also plan to use parameter-efficient fine-tuning (PEFT) methods such as LoRA to finetune a large model, directly producing the reverse-distilled embeddings at the last layer. This would facilitate generative use-cases of the model and enable likelihood- and logit-based probing methods. While our current approach introduces only a small-constant linear slowdown over the baseline methods, this approach would also provide a more efficient pipeline for downstream applications by requiring only a single forward pass, thereby completely erasing any slowdown.

---

> ### Author Response · Authors · 2025-12-03
> **Point by Point Responses -- Part 2**
>
> ## Reviewer nDBC
>
> ```
> The paper's primary weakness is the exclusion of the ESM-2 15B model. The most severe example of scaling failure is the performance degradation from 3B to 15B. The paper only demonstrates fixing the 650M-to-3B plateau. Without testing the 15B model, the central claim of “solving” the scaling paradox is incomplete.
> ```
>
> We have added the rd.15B parameter model to Tables 2 and 3, showing that we fix this most severe of scaling plateaus, and lending substantial evidence to the central claim of our manuscript.
>
> ```
> The core idea of using orthogonal subspaces to separate/disentangle knowledge, while novel in this application, is conceptually similar to methods in continual learning (e.g., O-LoRA), which should be cited.
> ```
>
> We have added the following text to the Related Work to address this oversight in discussion of continual learning.
>
> > Our approach is also related to, but distinct from, methods for continual learning like o-LoRA or Adaptive SVD. While these focus on maintaining model performance with adaptation to new tasks by ensuring orthogonal subspaces of weights are updated, our approach provides a task-agnostic framework for extracting representations from multi-scale model families by maximizing residual information gain.
>
> ## Reviewer xk7u
>
> ```
> The paper builds its entire motivation on the “general vs. specialized representation” hypothesis but does not provide a quantitative or qualitative analysis to validate it
> ```
>
> We have added quantitative support for our general vs. specialized representation hypothesis with our new SAE experiments in Figure 2/Section 3.4, where we show that rd.35M embeddings capture more specific sets of GO terms than baseline model embeddings. In addition, we have provided indirect support for this with our updates to Section 3.3 regarding performance on R2/R1 prediction.
>
> ```
> The proposed Optimal Constrained Approximation theorem only guarantees minimal reconstruction error under a prefix constraint: a standard property of linear least squares combined with SVD. However, this result does not theoretically justify why the assumed decomposition is needed or effective.
> ```
>
> We have added the following text to our Theoretical Analysis to justify the decomposition proposed by our method:
>
> > To enable scalability and flexibility in our representation space, it is desirable to enforce the Matryoshka property on our embeddings. Thus, we consider the set of all $k_p$-dimensional representations that preserve $M_r$'s embeddings in their first $k_r$ coordinates:}
>
> $$\mathcal{C}_r = \{[H_r, X] : H_r \in \mathcal{M}_r, X \in \mathbb{R}^{L \times (k_p - k_r)}\}$$
>
> > Our decomposition $H_{rd} = [H_r, H_{res}]$ minimizes reconstruction error within this constrained space
>
>
> ```
> Since RD performs a chain-wise representation enhancement, a straightforward baseline naturally arises: direct representation fusion across the same model chain. It remains unclear whether RD’s improvement comes from its “distillation mechanism” or simply from aggregating multi-scale features. A comparison against naive fusion, or fusion with simple KD objectives, is essential to establish the method’s actual contribution.
> ```
>
> We have evaluated a naive PCA-based test that performs comparably to our approach. However, we note that a PCA of fused embeddings will not satisfy the Matryoshka property, and thus does not support efficient scaling and training on prefixes of embeddings. A naive fusion of embeddings without PCA does not provide a direct comparison due to the increased embedding size and number of parameters. Our proposed distillation mechanism achieves aggregation of multi-scale features while maintaining scaling properties and compatibility with downstream prediction models.
>
> ```
> The experiments demonstrate improvement within the ESM-2 family, but it is uncertain whether the observed scaling restoration generalizes to other architectures.
> ```
>
> While we would like to expand reverse distillation to other foundation models, we consider it outside of the scope of the current work. We highlight the following text from our Limitations and Future Work:
>
> > Additionally, we will explore the effect of reverse distillation on other biological foundation models beyond ESM-2 to test the generalizability of our approach.

---

### Meta-Review · Area_Chair_WU8a · 2025-12-12

**Summary:**

All reviewers agree that the paper tackles an important problem: non-monotonic scaling in PLMs. They all find the proposed framework conceptually novel and empirically strong, but identify several concerns:
- C1. Reverse distillation (RD) experiments of 3B -> 15B are missing, where this non-monotonic phenomenon is the most severe.
- C2. "General vs specialized" representation hypothesis is not sufficiently supported.
- C3. The main theorem is a standard property and does not justify why the proposed decomposition is needed.
- C4. A direct comparison to simple representation fusion is required.
- C5. Generality beyond ESM-2 families is not studied.

**Reviewer Concerns:**

The authors' responses address the most critical concerns, particularly those regarding the 15B model and the representation hypothesis.
- C1. The rebuttal now includes results for rd.15B, showing that it outperforms both rd.3B and the base 15B model, and that it restores the expected monotonic scaling behavior.
- C2. The authors add two new experiments to support the representation hypothesis, which I find largely convincing.
- C3. The theoretical analysis is expanded to explicitly motivate the Matryoshka property (embeddings of larger models should contain smaller-model embeddings as a prefix). The SVD-based decomposition is justified as the optimal linear approximation within this constrained family.
- C4. The authors discuss the concatenation + PCA baseline and note comparable performance in some cases. While they emphasize that such fusion does not satisfy the Matryoshka property, I find these results interesting and recommend that they can be included in the revision.
- C5. The authors acknowledge this limitation and leave the exploration to future work. Although this does constrain the generality of the claims, I still view the contribution as significant given the central role of ESM-2 among current PLMs.

**Reviewer Scores:**

Overall, I think the rebuttal addresses the major concerns of all reviewers, and Reviewer xk7u would have increased the score from 4 to 6.

---

### Decision · Program_Chairs · 2026-01-26

Accept (Poster)